# Artificial Intelligence in Traditional Chinese Medicine: Past, Present and Future

Tian Xu
Shuguang Hospital Affiliated to
Shanghai University of Traditional
Chinese Medicine
Shanghai, China
22023284@shutcm.edu.cn

Liqiang Ni*
Shuguang Hospital Affiliated to
Shanghai University of Traditional
Chinese Medicine
Shanghai University of Traditional
Chinese Medicine Library
Shanghai, China
xyylc_nilq@126.com

Huaiqiong Zhang†
Shanghai Municipal Health
Commission
Shanghai, China
drzhanghq@139.com

## Abstract

Artificial intelligence (AI) technology has experienced rapid development, empowering various fields, including Traditional Chinese Medicine (TCM). This paper reviews the history, current status, and future prospects of AI applications in TCM. The integration of AI and TCM dates back to the 20th century, with the emergence of expert systems. However, their limitations led to a decline. In recent years, deep learning and natural language processing have enabled significant advancements in intelligent diagnosis, pattern recognition, and data mining of ancient medical texts. AI has been applied to TCM diagnostics, including tongue and facial diagnosis, pulse analysis, and assisted syndrome differentiation. Additionally, AI has facilitated drug development, medical education, and robotics in acupuncture and massage. Despite these achievements, challenges remain, such as the lack of standardized datasets, inadequate sample sizes, and interpretability issues. Concerns about ethics, data privacy, and potential job displacement have also arisen. To further harness the potential of AI in TCM, interdisciplinary collaboration, curriculum reform, and regulatory frameworks are crucial. By addressing these challenges, AI can revolutionize TCM's modernization, inheritance, and intelligent development while benefiting human health.

## CCS Concepts

• **Applied computing** → **Life and medical sciences**; **Education**;
• **Social and professional topics** → **Professional topics**.

## Keywords

Artificial Intelligence, Traditional Chinese Medicine, review

*Corresponding author
†Corresponding author

**Unpublished working draft. Not for distribution.**

**ACM Reference Format:**
Tian Xu, Liqiang Ni, and Huaiqiong Zhang. 2018. Artificial Intelligence in Traditional Chinese Medicine: Past, Present and Future. In *Proceedings of Make sure to enter the correct conference title from your rights confirmation emai (KDD-AIDSH 2024)*. ACM, New York, NY, USA, 9 pages. https://doi.org/XXXXXXX.XXXXXXX

## 1 Introduction

People probably never thought that the intelligent robot Eva depicted in the movie "Ex Machina", possessing human emotions and thoughts, would "appear" so quickly in our real world. On November 30, 2022, the intelligent chatbot ChatGPT developed by OpenAI was released to the public, attracting over a million registered users in just five days, and reaching 100 million active users within two months of its launch [1].

Artificial intelligence (AI) has now permeated various industries and sectors. Fields such as healthcare [2], education [3]architectural engineering [4], agriculture [5], and editorial writing [6] are facing transformations and challenges due to the rapidly evolving information technology.

AI refers to the ability of machines to think and act intelligently, similar to humans. It goes beyond merely replacing or simplifying human actions; it requires capabilities such as learning, reasoning, and simulation akin to human cognition. Since the concept of AI was introduced in the 20th century, research surrounding its applications in the medical field has been at the forefront. Early technologies were primarily applied in image recognition, statistical data analysis, and diagnostic assistance, achieving a series of significant results. As neural networks, deep mining, and natural language processing technologies made further breakthroughs, transformative advancements have been made in areas such as robotic surgery [7], medical image analysis, detection of drug interactions, identification of high-risk patients [8], and electronic health records [9].

The integration of TCM (TCM) and AI technologies, including but not limited to deep mining and understanding of classical ancient texts, standardization of TCM diagnostic and treatment data, and improving the ambiguity and irreproducibility of TCM syndrome differentiation, is believed to make resolving the long-standing issues of TCM inheritance and development more feasible. This integration can also provide robust support for TCM research, better promoting the modernization and intelligent development of TCM.

This article reviews the historical development of AI technology and relevant research in the medical field, particularly in TCM. It also elucidates and reflects on the current status and future trends of AI applications in TCM.

## 2 The development of AI technology in the field of medicine and TCM

### 2.1 Historical traceability

Before delving into the rapid development of AI, it is necessary to review its origins and the evolution of related technologies. Understanding this evolution allows us to better comprehend their significance in healthcare.

In 1954, Chinese-American scientist Professor Dr. Jiaqi Qian used computers to calculate dose distribution for radiotherapy, marking the first application of computers in medicine, though this was not yet considered AI technology.

In 1956, Professor McCarthy summarized and proposed the concept of AI at the Dartmouth academic conference in the United States [10].

In 1959, Professor Ledley from Georgetown University first applied Boolean Algebra and Bayes' Theorem to establish a computer-aided medical diagnosis model and successfully diagnosed a group of lung cancer cases [11]. This was the first attempt at medical diagnosis using AI technology.

In 1966, Professor Ledley formally proposed the concept of "Computer-Aided Diagnosis" (CAD)[11]. The first application that could be called a "chatbot" was born [12], the ELIZA chatbot system, a "psychological counseling doctor" developed by Professor Joseph Weizenbaum at MIT.

In 1968, the DENDRAL expert system for inferring chemical molecular structures was born[11]. Although it was not an application in the medical field, its emergence marked the birth of an important branch of AI – expert systems.

In 1972, de Dombal's acute abdominal pain differential diagnosis system was born [10]. In 304 cases, although both it and human doctors misdiagnosed some healthy cases, it completely diagnosed all cases of acute appendicitis, while human doctors had missed diagnoses. Moreover, its misdiagnosis rate was lower than that of human doctors.

In 1976, Professor Shortliffe at Stanford University developed the MYCIN medical expert system[10], which was based on a "rule-based" AI application. Due to the relatively complete forward and backward reasoning, and in-depth optimization search methods of the MYCIN medical expert system, later experts extracted its framework into the "EMYCIN" expert template, which was applied to various other expert systems. It became the technical foundation for various expert systems.

In 1979, the Guan Youbo Expert System for Diagnosing and Treating Liver Diseases was created, named after the renowned TCM physician Guan Youbo. This system marked the first expert system in TCM.The design concept of the TCM expert system was to first conduct medical theory design, combining the basic methods and characteristics of TCM pattern differentiation – the "four diagnostic methods", collect clinical symptoms into a database, then input the criteria for syndrome differentiation into the system to establish differentiation rules, collect clinical treatment methods for addition and subtraction, and at the same time establish rules for symptom-based addition and subtraction. Considering that different TCM experts have different reasoning paths for pattern differentiation, a specific set of knowledge base and rule base was established for this expert system tailored to that expert's knowledge. Subsequently, mathematical modeling was performed, and clinical testing and debugging were conducted. Since the reasoning rules of different medical experts were not identical, the logic of this TCM expert system was also based on the "methods methods" model, with the medical theory design part replaced by the different expert's knowledge base and rule base content, thus generating a personalized expert system.

In 1981, the Computer-Aided Diagnosis and Treatment System for Bi Syndrome was born, with a complete consistency rate of 96.88% [11].

In the 1990s, specialized expert systems for specific diseases emerged.

In 1995, the Chinese Acupuncture Teaching, Diagnosis and Treatment Expert System was demonstrated to experts and scholars from various countries at an international conference. This system simulated the acupuncture teaching of Professor Yu Zhishun and incorporated the clinical diagnosis and treatment thinking of renowned Chinese acupuncture experts at the time, representing an attempt to apply AI technology in the field of acupuncture.

From the 1980s to around 2010, there were more than 220 TCM expert systems. Although they experienced a heyday of entering the market and even being sold overseas, they ultimately declined and failed to achieve satisfactory results for doctors and patients. The main reasons for failure were outdated methodologies and models that did not effectively combine the reasoning and decision-making processes of pattern differentiation, leading to ineffective decision-making; a lack of cross-disciplinary talents, resulting in incomplete knowledge architectures of TCM in the knowledge bases, making it impossible to diagnose flexibly in practical applications; and the ambiguity of TCM pattern differentiation, where the lack of a unified and standardized knowledge architecture made it difficult to distinguish the uncertainty of diagnostic results by different doctors in clinical use [13, 14].

### 2.2 A Decade of Exploring and Comparing AI Algorithms in TCM

The period from around 2008 to 2017 marked a more comprehensive beginning of the integration of TCM with modern intelligent information technologies. If previous attempts mainly focused on clinical assisted diagnosis, then after 2008, efforts were made to improve the accuracy of the four diagnostic methods and develop and enhance multi-scenario robotic systems such as massage robots, intelligent prescription dispensing robots, and health monitoring robots.

In 2008, Fufeng Li et al.[15] conducted research on the computer-aided automatic recognition of facial complexion diagnosis in TCM by combining pattern recognition, AI, and other information processing technologies, achieving the standardization and precision of facial diagnosis.

In 2010, Huanbing Gao et al.[16] combined the TCM health massage expert system with multi-modal human-computer interaction

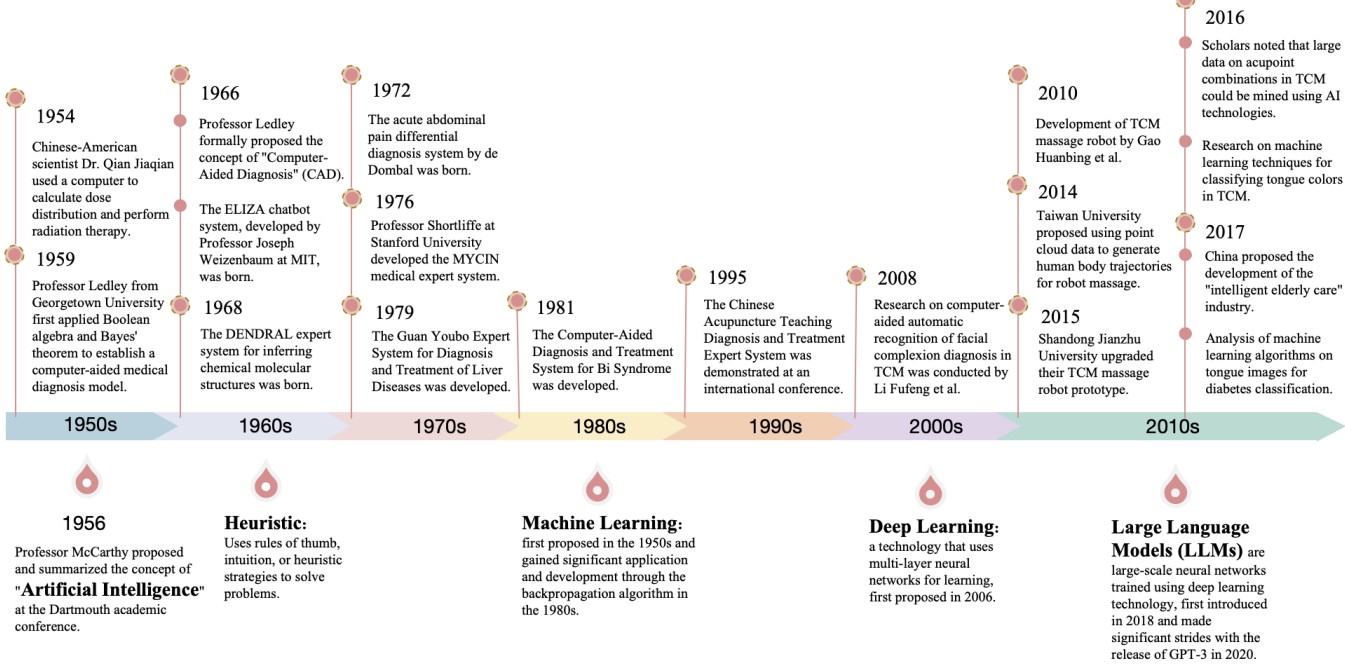

**Figure 1: AI Development Timeline in Traditional Chinese Medicine.**

control modules and other technologies in an attempt to develop a TCM massage robot. Mingjie Fan et al.[17] improved the deficiencies of traditional K-nearest neighbor and support vector machine classifiers in TCM facial diagnosis image processing, achieving technical enhancements in facial color feature extraction and accurate facial positioning. In the same year, Shandong Jianzhu University produced a prototype of a TCM massage robot and conducted product upgrades and series development in 2015 [18].

In 2014, National Taiwan University proposed using point cloud data to generate human body trajectories for robot massage [19].

In 2016, Chinese scholars noted that the large amount of data on acupoint combination in TCM could be data-mined using AI and other technologies to construct databases and explore the underlying logic of acupoint combinations, providing optimal treatment recommendations for clinical practice [20]. Qi Zhen et al.[21] explored the differences in performance of various machine learning techniques, such as random forests, SMOTE algorithms, and support vector machines, in classifying tongue colors in TCM tongue diagnosis, utilizing the strengths of each technique to improve classification accuracy.

In 2017, Jianfeng Zhang et al.[22] analyzed and compared the results of various machine learning algorithms on tongue images of diabetic patients, finding that the support vector machine algorithm achieved higher accuracy in establishing a classification model for diabetes. Also in 2017, China clearly proposed the development of the "intelligent elderly care" industry to address the issue of an aging population, aiming to realize the integration of "TCM +

intelligent elderly care" through intelligent monitoring devices and home service robots [23].

To facilitate understanding and observation, Figure 1 presents a timeline that organizes the historical evolution process.

## 3 The Current State of Rapid Development and High Integration of AI in TCM

Among the applications of AI in various industries, the medical field has consistently been a primary focus of research. In practical applications, the most mature category is image recognition, analysis, and prediction in radiology. This technology provides rapid image interpretation for clinical physicians, and has already become a reality in tasks such as breast cancer screening, ECG algorithms, and image processing for various ophthalmological retinal pathologies.

### 3.1 Image Analysis in TCM Visual Diagnosis

The accuracy of syndrome differentiation in TCM is based on the integration of the four diagnostic methods, with "observation" being the primary method. This includes observing complexion, body shape, posture, and tongue appearance. However, the process of "observation" is relatively subjective and influenced by both patient and physician individual behaviors. It is also limited by factors such as the clinical environment and natural conditions, which can interfere with the differentiation results. This is one of the reasons why traditional medicine has been unable to achieve

precision. Currently, the integration of AI technology with observational diagnosis mainly focuses on three aspects: eye diagnosis, tongue diagnosis, and facial diagnosis [24].

Traditional eye diagnosis in TCM is based on the "Five Wheels Theory", which infers the function of internal organs and the abundance of qi, blood, and body fluids by observing the shape, color, and condition of the eyes. Using smart wearable devices to collect real-time data from the eyes and face is the most direct and effective method. This approach can avoid the influence of human judgment and environmental factors. It also has practical significance for the accuracy of TCM syndrome differentiation and the establishment of personal data archives due to its ability to collect large amounts of data under different times and environments, as shown in the research by Zhu Huiming et al.[25]. Meanwhile, the imaging technology developed by Xue et al.[26] can bypass the influence of ambient light sources on eye observation. Several studies have employed AI techniques for auxiliary diagnosis of diseases through eye examination. For instance, Xiao et al.[27] conducted deep analysis of eye diagnosis data to achieve auxiliary diagnosis, primarily for liver and gallbladder diseases. Chen Xiuping et al.[28] developed an intelligent diagnostic system for syndrome identification through eye observation. This system combined morphological data of the scleral vessels with random forest technology to mine data characteristics of patients with blood stasis syndrome due to qi stagnation in the recovery period of stroke, providing precise quantitative predictions for syndrome types in the recovery period of stroke. If AI technology can be used to deeply mine eye diagnosis images and summarize the syndrome characteristics for various other diseases and conditions, the "Five Wheels Theory", which is not widely used in current TCM clinical practice, may have greater potential for development.

• Application of AI Technology in Tongue Diagnosis

Tongue diagnosis has been an important method for observing overall health in TCM for thousands of years. Yuan Li et al.[29] utilized machine learning to examine the value of tongue images and microbiome in gastric cancer diagnosis. Their research found that the diagnostic results were stable and significantly superior to traditional blood biomarkers. This breakthrough could potentially replace the long-standing reliance on invasive gastroscopy for diagnosing and screening gastric cancer, improving both public acceptance and operability, while also increasing the accuracy of pre-diagnosis for gastric cancer. The clinical application of this method appears highly promising. Similarly, Nan Jiang et al.[30] employed AI technology to conduct objective research on tongue appearance for the analysis of precancerous gastric lesions. Other scholars have approached this from a syndrome differentiation perspective, analyzing different tongue features under convolutional neural networks for specific sub-fields [31]. Some have even conducted objective analyses of tooth-marked tongues, a rarely studied area, using multi-layer neural network models [32]. These studies provide new insights for future research on the application of AI technology in tongue diagnosis.

• Application of AI Technology in Facial Diagnosis

There is relatively less research on facial diagnosis. In recent years, some scholars have conducted real-time analysis on the relationship between facial color, facial features, and diseases, mainly using

traditional AI algorithms [33]. Lin Yi et al. used six different convolutional neural network models to analyze and test facial images, achieving classification accuracy rates above 70%, significantly improving upon the effects of classification using traditional models [34].

## 3.2 Application of AI Technology in TCM Pulse Diagnosis Devices

Research on pulse diagnosis devices began quite early, and devices such as the ZM-III Intelligent Pulse Signal Instrument and the ZM-300 Pulse Signal Instrument are commonly used on the market [35], as well as the latest developed portable wireless pulse monitoring systems [36]. Although these devices are also referred to as "intelligent" instruments, hey actually only employ computer programming and signal communication systems, and do not truly employ "AI" applications. Some studies have utilized deep learning to establish pulse signal feature recognition methods. However, these approaches are more akin to signal processing and are not based on the principles of TCM syndrome differentiation [37]. However, in recent years, there has been some progress in research combining tongue and pulse information using AI technologies [38]. Domestic research groups have used a combination of unsupervised and supervised learning approaches in AI to analyze and model tongue and pulse information, studying the pulse diagnosis information characteristics of different groups such as pregnant women, as well as patients with hypertension, liver cirrhosis, chronic obstructive pulmonary disease, and fatty liver, significantly improving accuracy [39].

## 3.3 Intelligent consultation and auxiliary diagnosis

Although expert systems have long been a research focus of intelligent TCM, there has been no particularly obvious progress. Until the field of AI enters the era of deep learning, the repeated self-learning process of "input-learning-simulation-clinical trial-self-adjustment-repeated clinical trial-repeated self-adjustment" can be realized through deep learning, so as to improve the accuracy of AI syndrome differentiation and treatment. With the help of natural language processing technology, the in-depth study of TCM classic literature and expert encyclopedia knowledge base enables the computer to understand complex TCM medical records and promotes the development of intelligent consultation [40]. Chuanjie Xu's[41] research used pre-trained language models such as BERT to obtain TCM symptom text and prescription text without dosages. The analysis found that the TCM diagnostic prediction model using deep learning demonstrated good performance in pattern differentiation accuracy and herbal prescription recommendations. Yingjie Shi [42] conducted automated data collection through natural language processing techniques and data mining on a large amount of TCM diagnostic and treatment data, including syndrome names and pattern differentiation elements. For angina pectoris, he compared two pattern differentiation reasoning methods: one based on pattern differentiation elements and the other applying knowledge graph reasoning based on pathogenesis knowledge. Ultimately, he established an assisted diagnostic model integrating these two

pattern differentiation methods, providing insights for future intelligent assisted diagnosis in TCM. In Jianjun Lin [43] research, his mentioned that machine learning techniques have been widely applied in heart disease, breast cancer, and diabetes in China. He used data mining techniques such as Chinese word segmentation to process medical records and employed deep learning to extract yin-yang features from palm print images. He then used a BP neural network to establish a classification prediction model for TCM asthma syndrome differentiation. Yuhang Liu's [44] TD-TabNet model optimized the TCM asthma syndrome prediction system by integrating the advantages of different AI models. In the syndrome differentiation of hypertension [45] and the diagnostic decision-making of rheumatoid arthritis [46], research on assisted diagnosis has mainly utilized convolutional neural networks and other AI technologies to process facial image data and joint imaging data. Although multi-source information fusion intelligent algorithms can be applied not only to TCM clinical pattern differentiation but also to health management, their data analysis and deep mining capabilities can be utilized in multiple scenarios. Currently, the intelligent application of the four diagnostic methods in TCM is still in its initial stage in China [47]. Zhongren Sun et al. [48] mentioned, stating that due to the lack of classification standards and information differences, there is still substantial room for development in the application of AI technology to TCM pattern differentiation and treatment.

### 3.4 Electronic Medical Record Databases and Deep Learning

TCM medical records contain a large amount of unstructured data, and it is difficult to extract useful information using traditional techniques. Deep learning and natural language processing technologies can attempt to mine the knowledge information hidden within complex classical Chinese texts, which is of great significance for clinical decision support in electronic medical records [49–51].

Currently, Chinese researchers have used deep learning techniques to process unstructured data in electronic medical records, proposing natural language processing methods suitable for the characteristics of TCM outpatient electronic record texts. They have achieved precision in the yin-yang differentiation in clinical decision-making, syndrome differentiation of lung cancer and chronic gastritis, and recognition of TCM gynecological diseases such as polycystic ovary syndrome [48, 52].

### 3.5 AI Robot Assistance in Acupuncture and Massage

Current technology has essentially achieved precise acupoint localization. Examples include the TCM Qi-circulation and meridian-dredging robot [53], which utilizes robotic force control technology, 3D visual recognition technology, AI-assisted positioning technology, and voice interaction technology. Shanghai Jiao Tong University has developed a visual TCM massage robot [54] that combines deep learning technology to stably and efficiently obtain real-time human joint information. Nanchang University has constructed a mechanical arm for moxibustion on back acupoints [55] using HRNet and PFLD acupoint recognition deep learning technologies.

### 3.6 Data Mining of Ancient Medical Records and Database Establishment

TCM ancient texts are an important heritage of China's history and culture, serving as carriers of TCM cultural knowledge. By studying these texts, we can gain a comprehensive understanding of TCM's historical origins, theoretical systems, and clinical practice case records, enabling better inheritance and development of TCM culture. There are no fewer than 8,000 existing TCM ancient texts [56]. Digitizing these texts quickly and extracting their essence through systematic analysis is currently one of the key tasks in TCM heritage preservation.

Integrating AI technology can enable automatic identification and editing of variant characters, phonetic loan characters, errors, and inversions [57], and facilitate deeper mining, understanding, organization, and re-expression of ancient Chinese text. Beijing Jiaotong University has conducted research on named entity recognition and relationship extraction algorithms for ancient medical texts based on deep neural networks. These AI methods can rapidly and accurately extract medical knowledge from new literature, aiding in the construction of knowledge graphs for ancient medical texts [58]. In constructing knowledge graphs for TCM ancient texts, exploratory attempts have been made for postpartum abdominal pain [59] and jaundice [60] in "Synopsis of Golden Chamber", and for text information on abdominal distension in texts such as "Zhonghua Yidian" [61].

### 3.7 Pharmaceutical Manufacturing and Research Combined with AI Technology

Currently, there is an increasing demand for and quality requirements of Chinese herbal products. Utilizing modern technological means, aided by AI, network technology, big data, and other information technologies, is key to ensuring the quality of TCM preparations, improving medicinal safety, and promoting intelligent pharmaceutical research and development [62]. In drug research and development, by constructing multi-level biological networks, the phenotype-drug relationships underlying the intervention of herbal medicines in diseases and syndromes can be deeply analyzed, and their effectiveness can be verified. Based on AI, multi-modal multi-omics integrated analysis is advantageous for discovering key targets, active substances, synergistic effects of herbal components, and regulatory mechanisms within herbal formulas [63, 64]. Liansheng Qiao et al. [65] made innovative attempts in the development of new Chinese herbal medicines for treating chronic heart failure. Through targeted transcriptome sequencing and utilizing big data mining and AI computational methods, they identified pathways to reverse patients' dysregulated signals and intelligently screened Chinese herbal formulations.

### 3.8 The current status of utilizing AI technology in TCM education

Medical Education Medical education primarily encompasses two aspects: knowledge education and continuing education for medical students and healthcare professionals, as well as public health education for patients and the general public. Medical students hold a positive and optimistic attitude towards the emergence of

AI. In a survey of radiology students by dos Santos, DP et al. [66], most students believed that the integration of AI technology would significantly improve the field of radiology. Several universities abroad have begun to establish interdisciplinary courses combining AI and medicine, relying on medical knowledge, clinical experience, and digital expertise to solve modern medical and health problems [67]. In terms of patient and public education, the application of new technologies such as AI during the outbreak of the COVID-19 pandemic demonstrated effective responses. By systematically and rapidly collecting effective information from public platforms including social media and news media through big data screening, it not only assisted government decision-making but also encouraged medical professionals to continue voicing and providing accurate medical education on social media, which was a new and effective response to the epidemic event [68, 69]. Although many studies have shown that the accuracy of machine learning algorithms has surpassed that of expert doctors in many areas, and we know that combining AI technology in medical education can enable medical students to grasp medical knowledge more comprehensively and intuitively, medical education has not kept pace with the rapid development of AI. Currently, the design of TCM teaching courses integrating AI technology is still in the exploratory stage and has not been widely applied in clinical teaching [70].

## 4 The natural language processing capability of AI empowers the field of TCM

Chatbots, as a new intelligent tool based on natural language processing technology, capable of recognizing, understanding natural language, and interacting and expressing fluently based on context semantics, its widespread applicability is self-evident. Currently, major research teams and technology companies are strategically developing Chatbots projects, increasing investment in the research and development and technical enhancement of large language models. HuatuoGPT, developed by the Chinese University of Hong Kong, performed partially better than GPT3.5 in handling more complex multi-round consultation processes during evaluation testing [71]. Huawei's NLP Pangu medical large model can screen suitable drugs from massive compounds based on given targets or indications [72]. MedGPT by Medlinker, when tested against real doctors for diagnostic evaluation, achieved 96% consistency with the diagnostic scores of attending physicians from tertiary hospitals [73]. Large models specifically designed for TCM medical diagnosis are also under development.

## 5 Summary and Reflection
### 5.1 Current technical limitations and deficiencies of AI in the field of TCM

Although it has been more than half a century since the concept of AI was proposed, the birth of the TCM expert system in the mid-20th century, and the emergence of medical large language models today, the actual clinical application of AI technology in TCM still has many problems to be solved. The lack of standardized of TCM datasets is currently one of the biggest challenges. Due to the reasons such as the evolution of characters and the preservation of

ancient books over thousands of years of inheritance and continuation of TCM, it has increased the difficulty for subsequent research and development. Moreover, due to the complexity of the TCM knowledge system and the special nature of understanding the theoretical system based on feedback from practice, different schools of TCM have different diagnostic thinking and viewpoints, which is a manifestation of the richness and diversity of the TCM theoretical system. However, this particularity has also led to the lack of unified and standardized datasets, which greatly limits the construction of databases and the training and application of algorithms for AI technology. In addition, the lack of sample size is mentioned in most studies. Before the development of modern medicine and statistics, the data recording method of TCM was relatively simple, without systematic technical support and conditions, resulting in the loss of a large amount of important data. Furthermore, the clinical information of TCM involves a large amount of unstructured data, which brings great difficulties to information recording, collection, and sorting. At the same time, the lack of interpretability is also a problem in application. Since AI models are usually based on a large amount of data training and self-optimization, their decision-making process is "unknown," making it difficult to explain their reasoning process and logic at present. This "black box" algorithm leads to the possibility of significant deviations in the accuracy of its diagnostic results. At this point, experienced doctors need to check whether the results are correct through their own medical knowledge, but such requirements often exceed the capabilities of clinicians, which creates some interference for practical applications. At the same time, "uncertain" results may mislead patients or the public, which is not conducive to establishing a trusting doctor-patient relationship.

### 5.2 Concerns about the application of AI in the field of TCM

The application of AI in the field of TCM has also raised a series of concerns, mainly focusing on the following aspects.

- Research Integrity Issues

The rapid development of AI technology has made information acquisition easier, but it has also brought about issues of protecting original content and determining the authenticity of information. For example, in medical research, this could lead to data fabrication and the use of unauthorized medical databases or models, which is detrimental to the authenticity of research and its subsequent development.

- Medical Student Training Issues

While it has long been believed that virtual reality (VR) simulation technology in AI can achieve multi-dimensional, comprehensive presentation of knowledge, aiding in better understanding and experience, some studies have found that excessive immersive experiences may distract from learning, and the results are not necessarily better than traditional teaching methods. Additionally, medical students might not diligently master the necessary course knowledge or complete assignments through their own efforts, but instead overly rely on medical chatbots to do the work for them.

- Medical ethics issues

The ethical issues surrounding the application of AI in medicine have always been discussed, involving privacy, data security, accountability, and fairness. In the initial training phase, a large amount of personal health data needs to be collected and used, which involves personal privacy issues. If there are no relevant laws, regulations, and ethical norms to constrain and protect, data leakage and abuse may occur. In addition, when using AI systems for diagnosis and treatment, medical staff need to be responsible for the results, which requires users to have a strong knowledge base to determine the correctness of the results. However, due to the limited knowledge storage of the human brain and the "confusing" nature of the results, there is considerable difficulty in actual operation.

## 5.3 Exploring the future development direction of AI in the field of TCM

As countries such as the United States, Italy, and France have opened AI medical degrees and launched "medical engineering" courses and dual degrees such as Master of Medicine and Bachelor of Biomedical Engineering in recent years, if TCM wants to better tap the potential of AI technology, it can consider establishing dual degrees or more collaborative projects between medicine and computer science in medical education. For example, after studying TCM professional courses for two years in the undergraduate stage, students can add one year of learning the basics of AI in their upper grades, and then enter clinical practice training. As medical students, they do not need to spend time learning how to code in computer laboratories but can consider how to leverage the capabilities of AI algorithms and recognition in medical laboratories to better assist medical work. At the same time, after mastering certain AI technologies, it can better identify and match models and systems more suitable for clinical applications in specific departments, or propose technical requirements to computer professionals to achieve customized intelligent models.

When powerful chatbots appear, people start to worry whether they will be replaced by machines, and everyone will ask a question: "Which positions will be replaced by machines in the future?" Then, just like the industrial revolution that completed the transition from handicrafts to machine industry, this transformation not only changed the mode of production but also affected the adjustment of social structure and human lifestyles, but it brought human society into a better stage. In the current era of AI, we should also firmly believe that we can make better use of technology.

Currently, many research experiments are comparing the capabilities and accuracy between "AI medical models" and "human medical experts." When AI medical models outperform human medical experts in certain aspects, people start to worry. However, should we shift our focus to comparing the capabilities and accuracy between "human medical experts who can reasonably use AI medical models" and "human medical experts"? Perhaps this is the true meaning of the existence of AI technology as an auxiliary tool!

Medical workers should firmly believe that AI can facilitate and advance our medical work. In terms of helping doctors improve work efficiency, the use of technologies such as deep learning for assisted diagnosis can improve the diagnostic accuracy and speed of clinicians, reducing their workload. At the same time, it can assist

doctors in formulating personalized treatment plans, increasing patient satisfaction and treatment compliance, which can also help achieve better treatment effects and establish doctor-patient relationships. On the other hand, AI can help medical workers achieve better time allocation, using intelligent scheduling, online consultations, and other methods to rationally allocate consultation times. Especially for TCM clinicians, the four diagnostic methods and syndrome differentiation and treatment often require a longer time for each patient. By combining AI technology with the historical diagnostic and treatment thinking and methods of doctors, as well as considering the patient's past medical records and medical habits, it can analyze and reason out the medical consultation time for each patient, achieving more reasonable scheduling during appointment registration. Not only can it help save waiting time for patients, but it can also give doctors more rest time, improving work efficiency and quality of life. Additionally, by combining AI technology, medical workers can have more and better learning opportunities and methods, achieving the goal of lifelong learning, which will help improve the professional quality of medical workers and bring better medical services to patients.

In conclusion, while we focus on the integration of AI technology with TCM, we also need to pay attention to potential negative impacts, including ethical, legal, unemployment, and other issues. Governments, medical institutions, and medical schools should strengthen supervision by establishing corresponding regulatory measures. We believe that with the support of AI technology, TCM can better benefit human society.

## Acknowledgments

This study was supported by the Project supported by Shanghai Municipal Science and Technology Major Project (ZD2021CY001).

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

Received ; revised ; accepted
