# OpenReview forum: "Artificial Intelligence in Traditional Chinese Medicine: Past, Present and Future"
_KDD.org/2024/Workshop/AIDSH — KDD-AIDSH 2024 Poster_

### Official Review · Reviewer_AmQs · 2024-06-08
**A review about AI for Traditional Chinese Medicine**

**Rating:** 5
**Confidence:** 3

**Review:**

This review is about AI for Traditional Chinese Medicine, and authors show their insight in this field. However, in my opinion, this manuscript must be improved, as follows:
1. The typesetting throughout the document is subpar, markedly elevating the level of difficulty for the reader. It is suggested to employ a proper text alignment configuration or to utilize a more sophisticated writing tool.
2.  As a review paper, it is suggested to present the literature resource and numbers, and providing the reason why authors introduce and explain these literatures.
3. This paper do not show any figures or tables, making me confused. I strongly suggest authors supplement some tables and figures in the updated verison. For example, the history of this research field could be listed in a Table or a figure with timeline.
4. Please explain it:“Zhang Hongyi et al. proposed a method for pulse signal feature recognition using deep learning, but it is closer to signal processing and not based on the foundation of TCM pattern differentiation[34].”Reference[34]. Bi RY, Zhao YL, Zhu XL, Ma YH, Li JW, Zhang ZD, Xue CY. Research Progress on Digitalization of Pulse Diagnosis of Traditional Chinese Medicine[J]. Chinese Journal Of Sensors And Actuators. 2021;34(4):427-433. doi:10.3969 /j.issn.1004-1699.2021.04.001. Where is Zhang Hongyi? Authors are requested to carefully verify the citation of this article to avoid any similar issues.
5. The subtitle should be shorter, which could contains the essential information for this subsection.

---

### Official Review · Reviewer_LH1b · 2024-06-18
**Review：Artificial Intelligence in Traditional Chinese Medicine: Past, Present and Future**

**Rating:** 3
**Confidence:** 4

**Review:**

The authors claim that this manuscript reviews the history, current status, and future prospects of AI applications in TCM. However, the paper suffers from the following serious shortcomings:

1. Although it is a review paper, it still requires a clear motivation and rationale, along with a well-defined logical structure. It should provide readers with valuable insights or enhance their understanding. However, these qualities are lacking in this manuscript.
2. I feel the first section is not highly relevant to the main topic of this manuscript and could be further condensed and summarized.
3. What are the essential differences between image analysis of computer vision and the “observation” of TCM? How should their boundaries be defined? These concepts need to be clarified; otherwise, the concept of AI in TCM cannot be established.
4. The content in P4-5 is merely listed chronologically and requires a more in-depth explanation of the development logic of TCM.
5. The content in Section 3 is too brief and insufficient to support an entire section.
6. The manuscript lacks both figures and tables, which is not appropriate for an academic paper.

---

### Decision · Program_Chairs · 2024-06-28

Accept (Poster)